# The Changing Process of Women’s Smoking Status Triggered by Pregnancy

**DOI:** 10.3390/ijerph16224424

**Published:** 2019-11-12

**Authors:** Mai Itai, Akiko Sasaki, Makiko Mori, Shio Tsuda, Ayumi Matsumoto-Murakoso

**Affiliations:** 1Graduate School of Health Care Sciences, Tokyo Medical and Dental University, Tokyo 113-8510, Japan; sasaki.phn@tmd.ac.jp (A.S.); tsudphn@tmd.ac.jp (S.T.); 2Faculty of Health Care and Nursing, Juntendo University, Chiba 279-0023, Japan; 3Faculty of Nursing, National College of Nursing, Japan, Tokyo 204-8575, Japan; morima@adm.ncn.ac.jp; 4Treponte Children’s Clinic, Chiba 273-0867, Japan; bwalk75@gmail.com

**Keywords:** pregnancy, postpartum women, smoking, process, cessation, relapse

## Abstract

Although pregnancy is the trigger for many women to stop smoking, often they are unable to maintain cessation, undoing any health benefits for themselves and their children. Smoking is a complex phenomenon both before and after pregnancy, influenced by social background, relationships, and the specific experience of pregnancy and delivery. Therefore, it is necessary to clarify the experience and process of changes in women’s smoking status from pregnancy to after delivery. To explore possibilities for better smoking cessation support, the objective of this study was to clarify the changing process of smoking status from pregnancy to after delivery in women for whom pregnancy triggered a smoking cessation. We analyzed data obtained through semi-structured interviews with 31 women, using the grounded theory approach. Women reconsidered their smoking status, either quitting or smoking fewer cigarettes, because of externally motivated changes due to concerns regarding the influence of smoking on pregnancy and children. To prevent smoking relapse, it is important for the women themselves, as well as those around them, to appreciate their cessation, facilitating internal motivation and assessment of the situation. Furthermore, it is important to provide support, by implementing the process revealed in this study, not only during pregnancy but for an entire lifetime.

## 1. Introduction

Many studies have reported the harms that smoking among pregnant women can cause to both mother and child. Smoking in pregnant women has been shown to influence not only the fetus, resulting in low birth weight, preterm birth, and stillbirth, but also the occurrence of obesity [1] and asthma [2] later in children’s lives. Pregnancy and delivery can motivate women to stop smoking [3]. However, it has been shown that the failure rate of smoking cessation is greater, and the time within which smoking is resumed after a quit attempt is shorter, in women, as compared to men [4]. It has also been reported that women resume smoking while caring for children, despite having quit during pregnancy [5]. The smoking rate of pregnant women in Japan was respectively reported to be 2.5%–7.9% between 2006 and 2015 [5,6,7,8,9] and the cessation rate was 29%–83.8% between 2008 and 2013 [8,10,11,12]. Furthermore, the post-delivery rate of resuming smoking in women who stopped smoking during pregnancy was respectively reported to be 39.3%–70.3% between 2008 and 2013 [8,10,12,13,14].

In addition to the influence smoking during pregnancy has on the fetus, the effect of passive smoking on children who are in close contact with women who smoke is not negligible. Studies have shown that passive smoking in children can increase the risk of sudden infant death syndrome, respiratory disease, and arteriosclerotic disease [4], and that nicotine is transmitted to breast milk, affecting breastfeeding [15]. However, one study reported that some mothers believe that the harms of smoking can be prevented, as long as they do not smoke around their children, and the sense of being able to control their smoking can be especially strong immediately after delivery in women who stopped smoking during pregnancy [16]. It has been shown that women resume smoking after delivery because they believe it can no longer have a direct influence on their children, and the relationship between smoking and stress triggered by post-delivery fatigue and behavioral restrictions during childcare has been clarified [17]. Furthermore, smoking relapse after delivery has been shown to be related to various factors such as a young age, the smoking status of surrounding people, low income, and an early cessation of breastfeeding. However, it has been indicated that these barriers and facilitators are not fixed and mutually exclusive categories; instead, they are factors with a latent capacity to help or hinder smoking cessation [18]. It is, therefore, apparent that smoking in women is a complex phenomenon that can be influenced by social background, relationships, and the specific experiences of pregnancy and delivery. Accordingly, it is necessary to clarify the experiences and process of changes in women’s smoking status from pregnancy to after delivery. However, to our knowledge, no study so far has investigated this topic. Although pregnancy can trigger smoking cessation, many women cannot maintain this abstinence, resulting in smoking relapse and undoing any health benefits for themselves and their children. When we consider approaches to support smoking cessation and prevention of smoking relapse from pregnancy to after delivery, it is important to understand the changing process of smoking status in women for whom pregnancy was the trigger to stop smoking, their opinions on smoking and smoking cessation, and periods and situations where women have difficulty with smoking cessation.

Therefore, the objective of this study was to clarify the changing process of smoking status from pregnancy to delivery in women for whom pregnancy was the trigger to stop smoking, to help explore smoking cessation support for women.

## 2. Materials and Methods

We used the grounded theory approach [19] to clarify the changing process of women’s smoking status. This is a method to extrapolate concepts from data and to relate different concepts, which is appropriate for attempts to understand changing phenomena [20]. Therefore, it was ideal for our study, which aims to understand the changing process of smoking status in women who had experienced smoking cessation triggered by pregnancy.

### 2.1. Participants

The participants were women whose pregnancies had triggered smoking cessation and who were able to provide narratives about their experience and the process of changing smoking status from pregnancy to after delivery. The inclusion criteria were as follows: (1) aged 20 or older, (2) having been a smoker since before knowing about the pregnancy, (3) having stopped smoking for three months or more continuously, and (4) caring for a child aged three years or younger. In order to recruit a larger number of different participants there was no upper age limit for participants. However, to minimize recall bias, there was an inclusion of criterion (4) participants were caring for a child aged three years or younger. Participants were included regardless of smoking status at the time of the survey. However, we excluded those who had difficulties in (1) participating in the interview owing to psychological or physical problems and (2) responding to questionnaires or responding verbally in Japanese.

Recruitment at a pediatric clinic involved distributing handouts, displaying posters, and providing oral explanations of the study at the time of medical consultations. Candidates who were interested in participating in the research contacted the researcher using the supplied recruitment information. We gave detailed information about the research and obtained consent. The participants were all Japanese and recruited from one pediatric private clinic.

### 2.2. Data Collection

The data were collected by (1) a multiple-choice and self-administered questionnaire, composed of 15 questions covering participant characteristics such as age and working status, as well as smoking status-related factors including the age of smoking initiation and the number of cigarettes smoked in a day and (2) a semi-structured interview on the changes in smoking status to date and associated situations, feelings, and thoughts. The data were collected from March to July 2018 at the pediatric clinic or the participant’s home.

### 2.3. Data Analysis

The interview data were analyzed based on the Grounded Theory Approach. Transcripts were generated from the recorded interviews. After reading through the transcripts, the data were segregated and detached from context and then labeled using properties and dimensions. Labels were integrated to extrapolate categories by checking the similarities between labels and agreement between label properties. Then, the relationships between categories were examined. Structures were created by continuously comparing data in the process of analysis to capture the generated relationship between concepts in terms of both properties and dimensions. The data of 31 participants were analyzed by the method described above to understand the whole picture of the changing process of smoking status in women for whom pregnancy triggered smoking cessation.

### 2.4. Ethical Considerations

The research protocol was approved by the Institutional Review Board of Tokyo Medical and Dental University, Japan (approval number: M2017-293). Prior to conducting the investigation, we informed the participants of the research purpose and procedures and their rights in both verbal and written form. All participants provided written informed consent. We also obtained the participants’ permission to audio record the interviews.

### 2.5. Reliability and Validity

During the study period, our study group was supervised by researchers specialized in qualitative research, public health nursing, and psychiatric nursing. Furthermore, the validity of data interpretation and analyses were increased through regular supervision by researchers specialized in the grounded theory approach. In addition, the reliability of data interpretation and analyses were confirmed not only by study participants, but also by nursing and medical professionals with experience in providing support to pregnant women and mothers.

## 3. Results

### 3.1. Summary of Study Participants

This study had 31 participants, whose characteristics have been depicted in Table 1. The average age of the participants was 34.2 (25–42) years and the average number of children per woman was 1.8 (1–3) children. Regarding working status, 19 had a job (including maternity leave and childcare leave) and 12 did not. Regarding smoking status at the time of the survey, 18 participants (58%) were non-smokers and 13 (42%) were smokers. The average age of smoking initiation was 20.1 (14–23) years and the average number of cigarettes per day, or daily smoking status before pregnancy, ranged from smoking occasionally to 30 cigarettes per day. The time from delivery to smoking relapse in 13 present smokers was within one month for one participant, 2–6 months for eight participants, one year for three participants, and more than one year for one participant.

### 3.2. Changing Process of Smoking Status Triggered by Pregnancy

As per the analysis results, within the changing process of women’s smoking status triggered by pregnancy, we identified one core category, *objective review of present smoking status*, and eight subcategories. The following signs will hereby be used: { } for core category, ( ) for category, “ ” for category property, and [ ] for participants’ narratives.

#### 3.2.1. Whole Storyline

Women who started to (reconsider smoking triggered by pregnancy), whether by stimulating pregnancy or knowing about the pregnancy, either (smoked fewer cigarettes) or (initiated smoking cessation), depending on “the degree of recognition of the necessity for smoking cessation during pregnancy” or “the level of commitment toward smoking cessation”. Women who started smoking fewer cigarettes or stopped smoking entirely (dealt with the desire to smoke) through various “measures and approaches for smoking cessation”. When a low “degree of conviction to continue smoking cessation” and a high “level of desire to smoke” coexisted with no “measures and approaches for smoking cessation”, there was the possibility of unfavorable results. Many women, however, engaged in an {objective review of present smoking status} by effectively managing “periods when smoking cessation became difficult” and “factors that could increase the possibility of a smoking relapse”. Women generally tended to easily accept (support for smoking cessation) when the “level of self-evaluation of smoking cessation”, “level of others’ evaluation of smoking cessation”, and “level of recognition of merits of smoking cessation” were high. Women (maintained smoking cessation) when they had (support for smoking cessation) and “reasons to resist smoking relapse”. However, cases where the “level of self-evaluation for smoking cessation”, “level of others’ evaluation for smoking cessation”, and “level of recognition of the merits of smoking cessation” were low and when there was no (support for smoking cessation), women conducted an (evaluation of the stressful situation). (Smoking relapse) occurred when women considered it acceptable to smoke a few cigarettes when they were irritated, choosing to consider smoking a “countermeasure for stress”.

#### 3.2.2. Storylines for Each Theme

• Theme 1: Pregnancy as the trigger to reconsider smoking, leading to reduced cigarette intake or cessation

The “timing of reconsidering smoking” could correspond to various aspects of pregnancy, such as when women started to hope to be pregnant, during infertility treatment, when they knew they were pregnant, and if they had previously experienced a miscarriage. The following were identified as “opportunities that triggered the reconsideration of smoking”: the desire to have children, being aware of pregnancy, pregnancy at a very young or relatively old age, having a friend with a smoking habit who delivered a child with Down syndrome, and having information about the influence of smoking on the fetus from documents obtained from health centers.

Women had negative thoughts associated with smoking, such as a sense of guilt and concerns regarding the influence on children, which led to a lack of enjoyment in smoking. They obtained “knowledge about tobacco” using “sources of information” such as the internet, smartphones, TV, magazines, hospitals, and health centers. This increased “the degree of recognition of the necessity for smoking cessation during pregnancy” and “the level of commitment toward smoking cessation”, causing them to (initiate smoking cessation).

However, some women chose to (smoke fewer cigarettes) instead of quitting, for reasons such as not being able to give up smoking suddenly and believing that the associated stress is not good for the fetus. We found that the women had various “ways to smoke fewer cigarettes during pregnancy”, such as promising their husbands that they would smoke a limited number of cigarettes and smoking only the ends of their husbands’ cigarettes. Some, however, experienced a (smoking relapse) owing to anxiety regarding delivery and stress related to lifestyle changes caused by pregnancy. A woman who reduced her cigarette intake instead of quitting mentioned the following:


*[After six months of pregnancy, maybe because the pregnancy had been stable, I had the urge to smoke. However, owing to concerns about the baby, I promised my husband that I would not smoke more than half a cigarette at a time, and I kept my word. I did not smoke every day; I controlled myself as much as I could, but when the urge became unbearable, I gave in because being stressed is also not good for the baby.]*


In many cases, the “time to initiate smoking cessation” was when women knew they were pregnant. A woman who was a heavy smoker prior to her pregnancy had a low “level of difficulty in quitting smoking” as her “reason for smoking cessation” was concern regarding its influence on her child. In her own words:


*[I stopped smoking the day I knew I was pregnant. Although I was always aware of the harmful effects of tobacco, I used to smoke a lot. However, on learning of my pregnancy, my child’s life became my first priority and I was able to quit smoking right away.]*


The “confidence in the ability to quit smoking” was increased by physical reactions to smoking, such as morning sickness, and previous experience of smoking cessation triggered by pregnancy. On the contrary, being a heavy smoker and previous failed smoking cessation attempts decreased the “confidence in the ability to quit smoking”. Women engaged in various “ways to stop smoking”, which included (1) making a firm decision to quit without focusing on specific “ways to stop smoking”, (2) being patient, (3) autosuggestion, (4) making a declaration about quitting, and (5) being fined for smoking. In some cases, the “level of acceptance of smoking cessation” was high as pregnancy was considered a good opportunity for smoking cessation, while some “limited the period of smoking cessation” to the pregnancy or breastfeeding period. A woman who limited the period of smoking cessation to her pregnancy and resumed smoking after the delivery said the following:


*[If my smoking brings bad effect for baby, the baby needs to bear it for a whole life. So I need to push myself for a little and I was able to stop smoking as I thought I just needed to resist the urge for 10 months.]*


The “factors aiding smoking cessation” were as follows: (1) the discomfort caused by morning sickness, (2) making a joint effort with the husband to quit smoking, (3) having many friends who stopped smoking during pregnancy, and (4) feeling the movement of the fetus. A woman who used the picture of the fetus obtained during ultrasonography as a smoking cessation tool stated the following:


*[I saw what the ultrasonography image looked like after smoking; the fetus seemed to be suffering, and that probably helped me stop smoking. Also, I have heard that for the fetus, the mother smoking is equivalent to drinking water littered with tobacco; therefore, it was obvious that it is not a good thing for the baby.]*


• Theme 2: Objective review of present smoking status after dealing with the desire to smoke

The “level of desire to smoke” in women ranged from none to strong. They took various approaches to deal with the desire, such as not being alone, spending time with non-smoking friends or family, ceasing to buy tobacco, taking deep breaths, eating something when they felt the urge to smoke, looking up the harmful effects of smoking, and making an effort to recall the difficulty associated with smoking cessation. However, as a woman mentioned, *[smoking cessation requires a strong will]*, and there were many cases where they did not have specific “measures and approaches to manage the desire to smoke”.

It was apparent that the “period when smoking cessation became difficult” was when the women no longer experienced morning sickness, after they had delivered, at the conclusion of breastfeeding, when they returned to work, and when a child entered nursery school. One woman began to experience the desire to smoke immediately after delivery. At “times when the desire to smoke was particularly strong”, such as when they were away from children, when they were drinking alcohol, when they saw their family and friends smoking, and when they were faced with the same situations that would lead them to smoke in the past, they need to (deal with the desire to smoke). Furthermore, the “reasons increasing the possibility of a smoking relapse” were as follows: being surrounded by smokers, having cigarettes for the family in the house, and having no one to stop them from smoking. One woman told us about how seeing others smoking triggered within her a strong desire to smoke, as follows:


*[I never felt like smoking during morning sickness. But after morning sickness was over, I really wanted to smoke until the baby was born. My mother and husband are smokers, so every time I saw them smoking, I also wanted to smoke and I even had dreams of smoking.]*


Many women dealt with such difficult situations and demonstrated an {objective review of present smoking status}. Women recognized the merits of smoking cessation and performed self-evaluation for smoking cessation by “recognizing the influence of smoking on a child”. When the “level of self-evaluation for smoking cessation” was high, they (1) were satisfied with their present life after smoking cessation, (2) felt that it was good to quit smoking, (3) could be satisfied without smoking, (4) did not need to be conscious of stopping smoking, and (5) recognized the merits of smoking cessation for themselves.

Furthermore, women received “others’ evaluation for smoking cessation”. The “evaluators” were people around them, such as children, family, colleagues, and friends, or health care professionals. As part of “others’ evaluation for smoking cessation”, they were told that it was normal for them to stop smoking or that it was great that they stopped smoking, while some women said that they did not receive any evaluations.


*[We do not have many opportunities to be complimented. Since smoking is not something to be proud of to begin with, no one praised me for quitting smoking. So, it would be nice to have nurses who can praise me.]*


Furthermore, women had “opportunities to have their smoking status checked”, such as when they were issued the maternal and child health handbook, on home visits by public health nurses, and during workplace health checkups and general medical examinations. Although there are social trends against smoking, such as smoking being banned in the workplace, there is the possibility of the “surrounding environment leading to smoking”, such as living with a smoker or having friends who smoke despite having children.

Out of concern, some “surrounding smokers” did not smoke near pregnant women, while others asked them to join them when they smoked. Women sometimes had contradictory “thoughts related to the surrounding smokers”; while the smell of smoke could make them feel envious of smokers and make them desirous of smoking, they could also be bothered by the smoke and by the fact that they could not make others quit smoking.

Furthermore, (support for smoking cessation) included “supporters of smoking cessation” such as husbands, children, public health nurses, obstetricians, and gynecologists quitting smoking together; discussion regarding the anxiety related to smoking cessation; receiving support in housework; being taken out for a change of scene; and the people around them, such as their husbands and children, disliking smoking. The type of “people expected to be supporters” included husbands, nurses, and those who watched over them. The “reasons to not want to resume smoking” included recognition of the negative influences of smoking, the hope that their children would not smoke, a lack of enjoyment upon smoking for the first time after delivery, and the possibility of becoming pregnant in the future, leading to (maintained smoking cessation). Women spoke of being cautious of a relapse as follows: *[Since my child is a girl, I do not want her to smoke like me. I do not want to show her that I smoke]* and *[Now cigarettes are getting really expensive. My husband dislikes cigarettes]*.

While performing an {objective review of present smoking status}, when the “level of self-evaluation for smoking cessation” was low, the conclusion of delivery and breastfeeding lowered women’s resistance to smoking, and its apparent merits exceeded those of cessation. Some considered it acceptable to smoke, while some were caught between guilt and desire.


*[I should stop smoking forever. I felt it was very pity to start smoking again but I started to smoke because I wanted to be refreshed. I had a great sense of guilt, regretting that I smoked.]*


Furthermore, concerns over the influence of passive smoking on children led some women to use heated tobacco products after the delivery, in turn, reducing the sense of guilt over smoking. One woman stated the following:


*[I used to smoke tobacco before, and I remember hating myself for exposing my child to passive smoking. Now I smoke heated tobacco products, so my guilt because of smoking is lower.]*


• Theme 3: To evaluate stressful situations, either maintaining smoking cessation or relapsing

When the “level of self-evaluation for smoking cessation”, “level of others’ evaluation for smoking cessation”, and “level of recognition of the merits of smoking cessation” were low, the women performed (evaluations of stressful situations).

In addition to the stress of smoking cessation itself, the “stressful situations” included (1) the husbands failing to comprehend the difficulty involved in smoking cessation, (2) the husbands continuing to smoke while the women stopped smoking, and (3) the inability to feel refreshed because of the constraints on going out associated with taking care of children. Furthermore, arguments with husbands and older children being in a rebellious phase were also included among the “stressful situations”. When they had no other “countermeasures for stress”, some women took to smoking, or deemed it acceptable to smoke a few cigarettes when they were irritated, leading to (smoking relapse). However, some women coped by using effective “countermeasures for stress”, such as talking to their husbands or mothers, and by having a room in their minds for they do not think too much about smoking cessation. One woman told us her story of (maintaining smoking cessation) despite the difficulty involved, as follows:


*[There were many stressful occasions when I really wished to smoke. However, when I thought of the effort involved in trying to quit smoking again if I relapsed, I did not want to go through it all over again. Now I do not even want to smoke.]*


## 4. Discussion

This study was conducted to clarify the changing process of smoking status from pregnancy to after delivery in women for whom pregnancy triggered smoking cessation. From the results, (1) being vigilant about opportunities where women may reconsider smoking, and providing support for cessation, (2) supporting an objective review of present smoking status and (3) strengthening women’s ability to manage stress in order to support smoking cessation in society are important, when health care professionals support smoking cessation and to prevent smoking relapse from pregnancy to after delivery. We will discuss these three themes and support for women’s smoking cessation, based on the roles of health care professionals who assist women during pregnancy and childcare, as well as the present situation of the Japanese health and medical systems.

### 4.1. Being Vigilant about Opportunities for Women to Reconsider Smoking, and Providing Support for Cessation

As shown in previous studies [8,9,10,12,13], many women stopped smoking because of pregnancy, while reconsidering smoking, which is one step before the behavioral change of smoking cessation, was also triggered by pregnancy. Pregnancy is indeed a major life event, which brings about dramatic changes from the precontemplation stage to action stage in the transtheoretical model of behavior change [21]. Women for whom the behavioral change of smoking cessation was motivated by pregnancy without sufficient preparation were concerned about the influence of smoking on children and obtained knowledge about tobacco from sources such as the internet and TV, in addition to medical and health-related institutions or services. Therefore, it is important to provide support to enhance the level of recognition of the necessity of smoking cessation in women, using accurate information based on medical evidence in this phase, where women have an increased interest in smoking cessation. It is even possible to orient women who choose to smoke a reduced number of cigarettes toward smoking cessation by assessing their willpower regarding smoking cessation, providing knowledge about the harms of smoking, suggesting ways to cope with stress other than smoking, and intensive counseling. It is important to take multiple approaches to support smoking cessation instead of smoking a reduced number of cigarettes during pregnancy, which can trigger dramatic changes in the stages of behavioral change.

It has been shown that the motivation for smoking cessation among pregnant women is external, and that they view the changes as being imposed on them [22]. Participants in our study also “limited the period of smoking cessation”, such as quitting only during pregnancy or breastfeeding, showing that concerns regarding the direct influence of smoking on pregnancy and children were externally motivated. Therefore, it is suggested that women did not have adequate internal motivation to enhance the “level of acceptance of smoking cessation”. When they limited the period of smoking cessation, such as quitting only during pregnancy or breastfeeding, it was a goal set by themselves, which can be a strong motivator for a limited period, while strategic support is required to continue smoking cessation afterward. It is important for women to not only be externally motivated to quit smoking, such as because of the pregnancy and the existence of the child, but also to have internal motivation to improve their own health and lives. In order for women to stop smoking for themselves, it is important to provide support, so that they recognize the merits of smoking cessation in totality, in addition to recognizing the negative influences of smoking on pregnancy, delivery, and childcare.

During pregnancy and after delivery, women have various opportunities to interact with health care professionals. It is, therefore, important for these interactions to be used as a platform to provide support for smoking cessation. In Japan, based on the Maternal and Child Health law, women may be asked about their smoking history when they seek guidance during maternal health checkups, as well as during the interviews held at the time of issuing the Maternal and Child Health Handbook. Interviews regarding smoking history are an important first step in the provision of smoking cessation support. “It is said that successful intervention is primarily dependent on the identification of pregnant smokers, and that descriptive questions can eliminate the problem of non-disclosure associated with asking only “Do you smoke?” [23]. Furthermore, if women have already embarked on the smoking cessation journey after finding out about their pregnancy, they may say they do not smoke, leading to the assessment that there is no need for smoking cessation support. It is not appropriate to categorize people as non-smokers, or as having successfully quit, only because they do not smoke presently, as their support system may be inadequate. Women who attempt to stop smoking because of pregnancy should be viewed not as non-smokers, but as people undergoing a continuous process. It is necessary to check not only their present smoking status but also their smoking history, thoughts on smoking, and the possibility of smoking in the future. It is important to give health care professionals opportunities to monitor long-term smoking status to ensure the early detection of situations that make maintained smoking cessation difficult. Furthermore, even when women do not require immediate support, they can later experience difficulty with smoking cessation without having specific ways to stop smoking, as seen with our study participants. It is, therefore, important to provide long-term support, such as establishing specific measures for smoking cessation, making use of resources to support smoking cessation, demonstrating what women should do when they find it difficult to maintain smoking cessation, and making them aware that help is always available, should it be required.

### 4.2. Supporting Objective Review of Present Smoking Status

After changing their behaviors based on a reconsideration of smoking status, women objectively reviewed their present smoking status by dealing with the desire to smoke. There were various challenges involved in the process of managing the desire to smoke. In many cases, there were no specific countermeasures for the desire to smoke or approaches toward smoking cessation. In such cases, they tried to cope with the increased possibility of smoking and instances when the desire to smoke was the strongest. It would be important to enhance self-evaluation for smoking cessation through encouragement to perform a valid evaluation of the present situation and their self-care ability to deal with the desire to smoke.

Furthermore, in addition to self-evaluation, women were concerned about how their smoking and cessation were evaluated by others. They desired affirmation of their smoking cessation, expecting people watching over them to act as supporters. It has been shown that smoking cessation requires high self-efficacy [24]. Thus, it is important to provide support to maintain high self-efficacy over the entire period of smoking cessation, not only when pregnancy triggers smoking cessation. Health care professionals have regular opportunities to monitor women’s smoking status. At that time, however, it is necessary to ensure high self-efficacy among women by watching over them as supporters, rather than pressurizing them and bombarding them with information. Support from the people around them, in addition to effort and coping strategies, becomes particularly important when smoking cessation is challenging. As women may feel guilty about smoking, it is important for the people around them, including health care professionals, to provide support at times that are particularly difficult, as demonstrated by our study, rather than applying pressure by repeatedly checking their smoking status.

As in previous studies [5,6,10,12,13,14,25], the women’s narratives included the topic of their partners and families smoking, suggesting that it is important to provide smoking cessation support not only to women but also to surrounding smokers. In Japan, there are opportunities for group health education, such as parents’ classes, in health centers and hospital, where topics related to smoking are sometimes discussed. Along with the population approach, this type of occasion creates opportunities for a high-risk approach, that encourages both smoking fathers and mothers to reconsider their own smoking and provides information so that they can receive support if they decide to quit smoking. Women’s beliefs about smoking regarding the need for social support, especially from a partner, emerged as important [25]. When women’s partners smoke, they can be significant supporters if they work on smoking cessation together. However, when their partners do not smoke, the difficulty women experience in attempting to stop smoking cannot be easily understood. It is necessary to increase the number of “supporters for smoking cessation” by encouraging partners, family, and friends to understand the value of smoking cessation and to think about ways to support women’s smoking cessation. As women hesitated to start smoking again when their children disliked smoking, education on the importance of protecting children from smoke may not only prevent their own smoking, but also train smoking cessation supporters in society.

Furthermore, it is important for health care professionals to have up-to-date knowledge. Recently, an increasing number of smokers have begun using heated tobacco products or electronic cigarettes, requiring close monitoring of and attention to the emerging tobacco-related market [26]. The use of electronic cigarettes to quit smoking may be common in women of reproductive age, including those who are pregnant [27]. However electronic cigarettes were reported to induce toxicity, inflammation and oxidative stress in the mothers and can accumulate in the developing fetus, affecting intrauterine development [28]. In the present study, switching to heated tobacco products after delivery not only made it easier to return to smoking, but also reduced the sense of guilt associated with smoking. The increasing number of studies on heated tobacco products and electronic cigarettes have presented concerns regarding the passive smoking associated with the former, in addition to the influence on smokers. It is important for health care professionals to access these latest findings to provide accurate information about emerging tobacco-related products, including heated tobacco. Therefore, it is important to organize the environment and network resources so that supporters can access the necessary data.

### 4.3. Strengthening of Women’s Ability to Manage Stress to Support Smoking Cessation in Society

The word “stress” was mentioned by many of the women who experienced smoking relapses. The stressful situations identified in this study included not only those related to smoking, but also childcare and family relationships. A worsening of depressive and stress symptoms over postpartum was associated with an increased risk of smoking [29]. Women evaluated various stressful situations, and relapses were associated with the belief that smoking is a way to cope with stress. Health care professionals who interact with women through pregnancy, delivery, and childcare should identify potentially stressful situations and make assessments regarding the required support. While it is important to eliminate or at least minimize stress factors, sometimes it is difficult for supporters to directly solve the problems women encounter. It is, therefore, important to enhance women’s ability to manage stress, in order to develop coping strategies other than smoking. Support for women’s stress during childcare not only leads to smoking cessation, but also enhances their health and helps prevent child abuse. Even if women resist messages regarding the harms of smoking and do not accept cessation support, they may accept childcare support that is not specific to smoking cessation or help that seems unrelated to smoking cessation. Such support can indirectly help women refrain from tobacco use. A previous study reported a trend of women who discussed parenting on the internet being likely to resume smoking [8]. In addition to such web-based assistance, we believe it is important to provide support that enables women to express themselves and manage the stress associated with childcare. Furthermore, many facilities related to childcare prohibit smoking on the premises. Thus, visiting such facilities can be an opportunity for women to feel mentally refreshed, in addition to being in a situation where they physically cannot smoke. It has been shown that mothers who have experience of baby massage have less anxiety and stress about childcare, which are known to be factors related to child abuse [30]. Physical contact between mother and child through baby massage and exercise can not only reduce the mother’s childcare-related stress, influencing the relationship between mother and child, but also be an opportunity for smoking cessation, as women can spend time with their children without being concerned about the smell of smoke. However, there were women who told us that smoking is a better alternative to taking their frustrations out on their children. When sympathetic health care professionals listen to such experiences, they cannot strongly recommend smoking cessation, and are more tolerant of the women simply reducing their cigarette intake. Therefore, it is necessary for health care professionals to possess the capability to provide support without blaming smokers. Health care professionals need to understand the influence of smoking and must be able to judge when to provide support, and to what extent.

As providing smoking cessation support is difficult, health care professionals may feel helpless, faced with situations that may cause them to lose confidence in their ability to provide care. As women have numerous concerns other than their smoking habits throughout pregnancy, delivery, and childcare, smoking cessation support and enhancing women’s own health may not be given the highest priority. It would be optimal for supporters to be empowered to provide smoking cessation support with confidence. For this purpose, it is important for supporters to have information-gathering opportunities, such as workshops and lectures, and it is imperative to create a network of smoking cessation supporters. Furthermore, parental smoking is known to be a significant risk factor for allergies asthma and other lung diseases among children [31,32]. Questioning the family members of children who visit the pediatric department owing to allergic disorders about their smoking history can trigger awareness of the need for smoking cessation. Even when it is difficult for doctors and nurses in the pediatric department to provide smoking cessation support directly, they can still direct women to smoking cessation clinics or specialized institutions. We believe that it is possible to provide comprehensive support for initiating and maintaining smoking cessation by linking and making use of maternal and child health and smoking cessation support networks.

Furthermore, the population approach to smoking cessation is sometimes insufficient by itself. The difficulty of initiating and maintaining smoking cessation can be related to various individual and social high-risk factors, such as the relationship between smoking and psychiatric disorders, including substance use disorder and attention-deficit/hyperactivity disorder [33,34,35], vulnerability to stress, and a complex family environment. Health care professionals with specialized knowledge must play a role in screening for these high-risk cases. In particular, special treatment is required when smoking is related to tobacco-related disorders and psychiatric disorders. Thus, it is important to coordinate with and transfer relevant cases to the psychiatric department and other specialized institutions.

Therefore, various medical and public health institutions, such as the maternal and child health department, health and welfare department, psychiatric health department, obstetrics and gynecology department, pediatric department, psychiatry department, and smoking cessation clinics, are important resources for the initiation and maintenance of smoking cessation support in women. In order for women to start and continue smoking cessation, collaborative assistance, linking women, family, and supporting institutions, is essential.

### 4.4. Limitations and Future Directions

The data in this study were based on subjective narratives related to women’s memories of smoking-related situations. As the data were self-reported, the possibility of response bias cannot be eliminated. However, based on the objective of identifying the changing process of smoking status from pregnancy to after delivery, we tried to minimize the time since the associated event (i.e., recall bias) by only recruiting female participants caring for children aged three years or younger.

Furthermore, we recruited participants who provided consent to participate in our project from one study site. Although we attempted to sample women from diverse demographic, social economic, and family backgrounds, since smoking status and others’ reactions to smoking may be influenced by social and cultural background, the scope of the study could be further extended.

Additionally, we did not include women who continued smoking despite pregnancy. It can be expected that such women have a greater dependence on tobacco and experience more difficulty in smoking cessation as compared to our participants. Therefore, similar research in women with a high dependence on tobacco is required.

## 5. Conclusions

Our study clarified the changing process of women’s smoking cessation triggered by pregnancy. In order to ensure that women do not experience smoking relapses, it is important for them to have an appropriate, objective review of present smoking status, which enhances their internal motivation for smoking cessation. To accomplish this, it is important for health care professionals to (1) provide accurate information based on medical evidence, (2) take a positive approach to women’s cessation efforts and their self-care ability, (3) promote the utilization of social resources by assessing the periods and situations that make smoking cessation difficult, and (4) identify high-risk cases and refer them to specialized institutions.

Furthermore, it is important to support smoking cessation not only during pregnancy but for the entire lifespan by making use of the processes identified in this study.

## Figures and Tables

**Table 1 ijerph-16-04424-t001:** Characteristics of the participants (*N* = 31).

Characteristics	Items	*N*
Mother’s age	Mean ± SD (range)	34.2 ± 5.3 (25–42)
	≤25 years	3
	26–30 years	5
	31–35 years	8
	36–40 years	9
	≥41 years	6
Working status	Working	19
	Not working	12
Final education	High school	15
	Junior college/technical college	9
	Undergraduate, postgraduate	7
Number of children	1	12
	2	14
	3	5
Present smoking status	Smoker	13 (41.9%)
	Non-smoker	18 (58.1%)
Age at smoking onset	≤14 years	1
	15–19 years	3
	20 years	20
	≥21 years	7
Have opportunity for interview about smoking history during pregnancy		
	Yes	26 (83.9%)
	No	5 (16.1%)
Have opportunity for interview about Smoking history after delivery		
	Yes	12 (38.7%)
	No	19 (61.3%)

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
