# Peer review of "The Changing Process of Women’s Smoking Status Triggered by Pregnancy"

_ijerph, 2019, doi:10.3390/ijerph16224424_

Round 1
Reviewer 1 Report
This is a well-written manuscript presenting findings from qualitative research on a significant public health issue. The study is novel and the findings are new and relevant from the point of view of healthcare professionals dealing with smoking cessation. There are some minor remarks that would need corrections to improve the quality of the paper: 1) Materials and Methods lines 75-77. Please provide a more detailed description of the participants. Where are they from? How participants were recruited? Are they from one clinic or more? Are they from private or public entities (or both)? Please specify. 2) Discussion lines 294-297. Please re-write those sentences. Usually, the first paragraph of the discussion should summarize the main findings, which will be discussed in the following paragraphs. The current version of the first paragraph in the Discussion section is unusual and needs improvement. 3) Discussion lines 375-377. Please provide references when citing other studies. Eg. “As in previous studies [], the women’s narratives included the topic of their partners and families smoking, suggesting that it is important to provide smoking cessation support not only to women but also surrounding smokers." 4) Conclusions. This section is too extensive and inconclusive. Please consider limiting the conclusions to a few sentences. Please prepare more comprehensive conclusions, which will be related to the main findings of the study. The current version does not provide this.Author Response
Thank you for inviting us to submit a revised draft of our manuscript entitled, “The Changing Process of Women’s Smoking Status Triggered by Pregnancy” to International Journal of Environmental Research and Public Health. We also appreciate the time and effort you have dedicated to providing insightful feedback on ways to strengthen our paper. Thus, it is with great pleasure that we resubmit our article for further consideration. We have incorporated changes that reflect the detailed suggestions you have graciously provided. We also hope that our edits and the responses we provide satisfactorily address all the issues and concerns you have noted.
To facilitate your review of our revisions, the following is a point-by-point response to the comments and questions.
Materials and Methods lines 75-77. Please provide a more detailed description of the participants. Where are they from? How participants were recruited? Are they from one clinic or more? Are they from private or public entities (or both)? Please specify.
RESPONSE: Thank you for the suggestion. We add the sentence “The participants were all Japanese, and recruited from one pediatric private clinic.” in line 84-85.
Discussion lines 294-297. Please re-write those sentences. Usually, the first paragraph of the discussion should summarize the main findings, which will be discussed in the following paragraphs. The current version of the first paragraph in the Discussion section is unusual and needs improvement.
RESPONSE: Thank you for the suggestion. We added and changed to the following sentences in the first paragraph of the discussion.
“This study was conducted to clarify the changing process of smoking status from pregnancy to after delivery in women among whom pregnancy triggered smoking cessation. From the results, 1) being vigilant about opportunities for women to reconsider smoking, and providing support for cessation, 2) supporting objective review of present smoking status and 3) strengthening of women’s ability to manage stress to support smoking cessation in society are important when health care professionals approach to support smoking cessation and prevention of smoking relapse from pregnancy to after delivery.”
Discussion lines 375-377. Please provide references when citing other studies. Eg. “As in previous studies [], the women’s narratives included the topic of their partners and families smoking, suggesting that it is important to provide smoking cessation support not only to women but also surrounding smokers."
RESPONSE: Thank you for the suggestion. We missed the reference, and added specific reference there.
Conclusions. This section is too extensive and inconclusive. Please consider limiting the conclusions to a few sentences. Please prepare more comprehensive conclusions, which will be related to the main findings of the study. The current version does not provide this.
RESPONSE: Thank you for the suggestion. We revised and clarified the conclusion in line 502-511.
Again, thank you for giving us the opportunity to strengthen our manuscript with your valuable comments. We had worked hard to incorporate your feedback and hope that these revisions persuade you to accept our submission.
Sincerely,
Mai ITAI

Reviewer 2 Report
The upper age limit of the study participants is not included for the study, the number of study participants is very less and the study area is not described.
The cause of the smoking status and changing process during pregnancy is already know. The conclusion drawn from the overall research is not sufficient.
Author Response
Thank you for inviting us to submit a revised draft of our manuscript entitled, “The Changing Process of Women’s Smoking Status Triggered by Pregnancy” to International Journal of Environmental Research and Public Health. We also appreciate the time and effort you have dedicated to providing insightful feedback on ways to strengthen our paper. Thus, it is with great pleasure that we resubmit our article for further consideration. We also hope that our edits and the responses we provide satisfactorily address all the issues and concerns you have noted.
RESPONSES: Thank you very much for your fair assessment and valuable comments.
In this study, we didn’t have upper age limit of participants, however we have inclusion criteria such as caring for a child aged three years or younger to minimize recall bias. Our study was qualitative research and the number of participants were limited as you mentioned, further research should be approach to more participants and another research design will be considered. In our study, we would like to clarify the changing process of smoking status not only during pregnancy but also from pregnancy to after delivery because there are a lot of mothers who relapse. We have redrafted the conclusion section in line 503-512 to establish a clearer focus. We believe that our result of this process can be referred any timing during pregnancy to after delivery and help health care professionals to understand the process and support mother’s smoking cessation process.
Again, thank you for giving us the opportunity to strengthen our manuscript with your valuable comments. We had worked hard to incorporate your feedback and hope that these revisions persuade you to accept our submission.
Sincerely,
Mai ITAI

Reviewer 3 Report
This is a descriptive study of smoking cessation and limitations among pregnant women and young mothers in a clinical setting. The study has the potential to contribute to the literature. However, there are some limitations that should be enhanced before this paper can be published.
The aim of this study was to clarify the changing process of smoking status from pregnancy to after delivery in women among whom pregnancy triggered smoking cessation.
Women had negative thoughts associated with smoking, such as a sense of guilt and concerns regarding the influence on the fetus, which led to a lack of enjoyment in smoking. The health concerns were built on strong desire to have a child (either after infertility treatment or natural pregnancy at a very young or relatively old age) and on the information from different sources i.e. health centers, friends, mass media. However, some women chose to smoke fewer cigarettes instead of total smoking cessation, for reasons such as not being able to give up smoking instantly and believing that the associated stress is not suitable for the unborn child. In some cases, the smoking cessation rate was high as pregnancy was considered a good opportunity for smoking cessation, while some women limited the period of smoking cessation to the pregnancy or breastfeeding period only. In general, the factors helping women smoking cessation were as follows: 1) the discomfort caused by morning sickness, 2) making a joint effort with the husband to quit smoking, 3) having many friends who stopped smoking during pregnancy, and 4) feeling the movement of the fetus. It seemed in most cases that women considered their unborn child’s health as number one priority. A factor that helped women keep not smoking after delivery was family’s and friends’ enthusiasm and compliments on not smoking. When exposed to stressful situation or if the husband remained a smoker after child’s birth, some women restarted smoking, or deemed it acceptable to smoke a few cigarettes when they were irritated, leading to smoking relapse. Pregnancy was the trigger for women to reconsider smoking, and they either smoked fewer cigarettes or initiated smoking cessation. Their motivations were either internal or external (external being much more common), based on concerns for the direct and indirect influence of smoking on pregnancy and children.
Strengths:
-Regular supervision by researchers specialized in the grounded theory approach makes the data more reliable.
-The authors visualized that it is essential to provide support to enhance the level of recognition of the necessity of smoking cessation in women using accurate information based on medical evidence in this phase (many women still rely on friends’ opinion and mass media information), where women have increased interest in smoking cessation. The authors believe that it is possible to adjust women who choose to smoke a reduced number of cigarettes toward smoking cessation by assessing their willpower regarding smoking cessation, providing knowledge about the harms of smoking, suggesting ways to cope with stress other than smoking, and intensive counseling.
-This article underlines the need to perform a better ‘smoking cessation counseling’ for pregnant women and describes that even though women have a lot of contact with medical services throughout pregnancy, the question ‘Do you smoke” is insufficient, and not much is provided when given the positive answer.
-Minimalizing the recall bias by only recruiting female participants having children aged three years of age or younger.
Weaknesses:
-The group of participants was 31 women, which rather should not be considered a representative group- that’s why the authors used the grounded theory approach. Still, some more specific data are missing.
- The lack of percentage in answers makes the article less transparent. I would be very interested in getting to know response rates.
- Prior to conducting the questionnaire, the authors informed the participants of the research purpose which might have influenced participants’ answers. The participants might have hidden the truth from the interviewer. The response bias is relatively high.
- Women with high dependence on tobacco were excluded from the study which might have an impact on the results.
- The authors didn’t write strengths of their study. It is the first study to evaluate smoking cessation during pregnancy which should be highlighted by the authors.
Specific comments
Line 37 “…the cessation rate was 29–83.8% between 2008 and 2013”
This sentence is unclear. Is it 29% in 2008 and 83.8% in 2013, and the word RESPECTIVELY is missing? If not, why were there such vast differences in rates reported?
Line 38-39 “…the post-delivery rate of resuming smoking in women who stopped smoking during pregnancy was reported to be 39.3–70.3% between 2008 and 2013”
The same questionable data here.
Line 74 Participants
The information if authors considered which pregnancy for the woman it was was taken into account is missing. Have the authors asked about it, and maybe are there any results suggesting that the first pregnancy was a stronger trigger for quitting smoking then following pregnancies? The results of this question could lead to some interesting conclusions, i.e., it is crucial to carefully interview women about their smoking habits in their first pregnancy.
Line 85 Data collection
What type of questionnaire was it? Was it a multiple-choice, or were there just spaces left to be filled out by participants? Who literally filled out the survey? How many interviewers were there? How many questions did the questionnaire consist of?
Line 131-150 Whole storyline
This paragraph is not entirely clear to me. It should be rewritten in a more accessible form.
Line 270-272 “Furthermore, concerns over the influence of passive smoking on children led some women to 270 use heated tobacco products after the delivery, in turn, reducing the sense of guilt over smoking.”
This would be very interesting to investigate more. How many women switched to heated tobacco products (HTP)? Was the approach towards heated tobacco products taken into consideration in the questionnaires? The data about starting HTP is missing. Did women consider switching to HTP as quitting smoking or limiting cigarettes? This issue should have been more explored as it is a very up-to-date topic.
Line 391-402 ”Furthermore, it is important for health care professionals to have up-to-date knowledge. Recently, an increasing number of smokers have begun using heated tobacco products or electronic cigarettes, requiring close monitoring of and attention to the emerging tobacco-related market…”
Is there an evidence-based-medicine confirmed effect of heated tobacco products on fetus and placenta? The authors should consider putting in some of the information about the influence of electronic cigarettes on the unborn child.
Line 443-444 “Furthermore, parental smoking is known to be a significant risk factor for allergies among children.”
It is important to highlight that tobacco smoke enhances not only allergies but also asthma and other lung diseases. To address this topic, the following article should be taken into account as it describes in detail the complex process:
Strzelak A, Ratajczak A, Adamiec A, Feleszko W. Tobacco Smoke Induces and Alters Immune Responses in the Lung Triggering Inflammation, Allergy, Asthma and Other Lung Diseases: A Mechanistic Review. Int J Environ Res Public Health. 2018;15(5):1033. Published 2018 May 21. doi:10.3390/ijerph15051033
Author Response
Thank you for inviting us to submit a revised draft of our manuscript entitled, “The Changing Process of Women’s Smoking Status Triggered by Pregnancy” to International Journal of Environmental Research and Public Health. We also appreciate the time and effort you have dedicated to providing insightful feedback on ways to strengthen our paper. Thus, it is with great pleasure that we resubmit our article for further consideration. We have incorporated changes that reflect the detailed suggestions you have graciously provided. We also hope that our edits and the responses we provide satisfactorily address all the issues and concerns you have noted.
To facilitate your review of our revisions, the following is a point-by-point response to your specific comments and questions.
Line 37 “…the cessation rate was 29–83.8% between 2008 and 2013”
This sentence is unclear. Is it 29% in 2008 and 83.8% in 2013, and the word RESPECTIVELY is missing? If not, why were there such vast differences in rates reported?
RESPONSE: Thank you for this suggestion. We add the word “respectively” in line 36. The differences were probably caused not only conducted year of survey but also the timing of survey (eg. Participant’s week of pregnancy and how long after delivery), the number of participants and so on.
Line 38-39 “…the post-delivery rate of resuming smoking in women who stopped smoking during pregnancy was reported to be 39.3–70.3% between 2008 and 2013”
The same questionable data here.
RESPONSE: Thank you for this suggestion. We add the word “respectively” in line 39.
Line 74 Participants The information if authors considered which pregnancy for the woman it was taken into account is missing. Have the authors asked about it, and maybe are there any results suggesting that the first pregnancy was a stronger trigger for quitting smoking then following pregnancies? The results of this question could lead to some interesting conclusions, i.e., it is crucial to carefully interview women about their smoking habits in their first pregnancy.
RESPONSE: Thank you for your question and suggestion. One of our inclusion criteria is caring for a child aged three years or younger. As shown in the table1, there are some mothers who have two or more children however there are no data that will indicate that first pregnancy was a stronger trigger for quitting smoking.
Line 85 Data collection What type of questionnaire was it? Was it a multiple-choice, or were there just spaces left to be filled out by participants? Who literally filled out the survey? How many interviewers were there? How many questions did the questionnaire consist of?
RESPONSE: Thank you for your questions. We used a multiple-choice and self-administered questionnaire, and it was composed 15 questions in 6 section and add this sentence in line 87-88.
Line 131-150 Whole storyline This paragraph is not entirely clear to me. It should be rewritten in a more accessible form.
RESPONSE: Thank you for your comment. In this section we try to introduce all the terms of subcategories and core category. In addition, this paragraph shows the general stories and the relationship between concepts were described using category property so that the sentences may not natural and clear. However, we would like to mention detailed storylines in next section. It may be thought that a storylines are selected arbitrarily, if we describe storylines for each theme only. Therefore, we think that it is necessary to first describe the summary of the story, introduce the entire story indicated by the interview data, and then describe the details of the selected story in particular. We hope this response helps you to accept our thought.
Line 270-272 “Furthermore, concerns over the influence of passive smoking on children led some women to use heated tobacco products after the delivery, in turn, reducing the sense of guilt over smoking.” This would be very interesting to investigate more. How many women switched to heated tobacco products (HTP)? Was the approach towards heated tobacco products taken into consideration in the questionnaires? The data about starting HTP is missing. Did women consider switching to HTP as quitting smoking or limiting cigarettes? This issue should have been more explored as it is a very up-to-date topic.
REPONSE: We totally agree with your comment that heated tobacco products issue should be more explored. When planning this study, we didn’t consider this heated tobacco product and didn’t ask for all participants in the questionnaires however some participants mentioned about this issue in interview. The data related heated tobacco products was limited in this study. We are planning to have another research related this issue.
Line 391-402 ”Furthermore, it is important for health care professionals to have up-to-date knowledge. Recently, an increasing number of smokers have begun using heated tobacco products or electronic cigarettes, requiring close monitoring of and attention to the emerging tobacco-related market…” Is there an evidence-based-medicine confirmed effect of heated tobacco products on fetus and placenta? The authors should consider putting in some of the information about the influence of electronic cigarettes on the unborn child.
RESPONSE: Thank you for the suggestion. There are limited research that considering the new tobacco products. We added a reference that are mentioned about the influence of electronic cigarettes on the unborn child.
Line 443-444 “Furthermore, parental smoking is known to be a significant risk factor for allergies among children.” It is important to highlight that tobacco smoke enhances not only allergies but also asthma and other lung diseases. To address this topic, the following article should be taken into account as it describes in detail the complex process: Strzelak A, Ratajczak A, Adamiec A, Feleszko W. Tobacco Smoke Induces and Alters Immune Responses in the Lung Triggering Inflammation, Allergy, Asthma and Other Lung Diseases: A Mechanistic Review. Int J Environ Res Public Health. 2018;15(5):1033. Published 2018 May 21. doi:10.3390/ijerph15051033
RESPONSE: Thank you for introducing important paper. We added the reference in line 455.
Again, thank you for giving us the opportunity to strengthen our manuscript with your valuable comments. We had worked hard to incorporate your feedback and hope that these revisions persuade you to accept our submission.
Sincerely,
Mai ITAI

Round 2
Reviewer 2 Report
The following concerns should be addressed,
Line 70: The specific reason behind not specifying the upper age limit should be explained in the letter.
Line 83: how the participants Recruitment done? It should be explained step by step.
Line 84-85: The participants were all Japanese and recruited from one pediatric private clinic. Be clear about the pediatric private clinic such as name, place, type of clinic and its description. Author should be very clear about the description.
Line 87: multiple-choice and self-administered questionnaire. The multiple-choice and self-administered questionnaire should be attached as supplementary.
Line 91-92: The data were collected between March and July 2018 at the pediatric clinic or the participant’s home. Is the study conducted from March to July or in the month of march and july? It should be clarified. Is your study is hospital based or community based?
Line 103: Ethical Considerations. The ethical clearance to conduct the study was obtained from the University. How you have obtained permission to conduct the study in the specific hospital or clinic? How this permission was obtained?
Overall, the manuscript should be edited for further clarity.
Author Response
We wish to thank the reviewer for insightful feedback and comment. We also hope that our edits and the responses we provide satisfactorily address your concerns. To facilitate your review of our revisions, the following is a point-by-point response to your specific comments and questions.
Line 70: The specific reason behind not specifying the upper age limit should be explained in the letter.
RESPONSES: Thank you for this suggestion. We add the following sentences in line 79-82. In order to recruit a larger number of different participants there was no upper age limit for participants. However, to minimize recall bias, there was an inclusion of criterion 4) participants were caring for a child aged three years or younger.
Line 83: how the participants Recruitment done? It should be explained step by step
RESPONSES: Thank you for your question. We add the following sentences in line 87-89. Candidates who were interested in participating in the research contacted the researcher using the supplied recruitment information. We gave detailed information about the research and obtained consent.
Line 84-85: The participants were all Japanese and recruited from one pediatric private clinic. Be clear about the pediatric private clinic such as name, place, type of clinic and its description. Author should be very clear about the description.
RESPONSES: Thank you for your suggestion. We didn’t specify the clinic name and place, because participants were patient in this clinic. We thought that revealing detailed information about the clinic bring ethical issue. We should not to disclose any information that could identify an individual. We hope this response helps you to accept our thought.
Line 87: multiple-choice and self-administered questionnaire. The multiple-choice and self-administered questionnaire should be attached as supplementary.
RESPONSES: Thank you for your suggestion. This is a descriptive study. We selected not to attached this questionnaire because questions were just about participants characteristics such as age, working status, number of children, children age as well as smoking status. After getting consent from participants, participants filled in this questionnaire and were interviewed. We selected this data collection procedure, because questionnaire can easily collect demographic characteristics data. In addition, the time of filling in this questionnaire helped participants to relaxing before interview. It was our pleasure to express our thought about this issue. We hope you accept our thoughts.
Line 91-92: The data were collected between March and July 2018 at the pediatric clinic or the participant’s home. Is the study conducted from March to July or in the month of march and july? It should be clarified. Is your study is hospital based or community based?
RESPONSES: Thank you for your question. We conducted data collection from March to July. We changed this in line 93. In addition, our study was not only hospital based but also community based. In Japan, we have a national health insurance system and all citizens from the poor to the rich have health insurance. This private clinic can be accessed from neighborhood with all insurance. We thought that our study can be said both hospital based and community-based.
Line 103: Ethical Considerations. The ethical clearance to conduct the study was obtained from the University. How you have obtained permission to conduct the study in the specific hospital or clinic? How this permission was obtained?
RESPONSES: Thank you for your question. Our research cooperation clinic doesn’t have an ethical review board. We received an ethical review consignment form from director of the clinic. All research protocol include ethical review consignment from the clinic was approved by the Institutional Review Board of Tokyo Medical and Dental University, Japan (approval number: M2017-293).
In addition, Our manuscript had checked by professional English editing service.
Again, thank you for giving us the opportunity to strengthen our manuscript with your valuable comments. We had worked hard to incorporate your feedback and hope that these revisions persuade you to accept our submission.
Sincerely,
Mai ITAI
